# Myocardial Edema: A Rare but Specific Manifestation of Neonatal Capillary Leak Syndrome

**DOI:** 10.3390/diagnostics13233597

**Published:** 2023-12-04

**Authors:** Jing Liu, Yue-Qiao Gao

**Affiliations:** 1Department of Neonatology and NICU, Beijing Obstetrics and Gynecology Hospital, Capital Medical University, Beijing 100026, China; 2Department of Neonatology and NICU, Beijing Chao-Yang Hospital, Capital Medical University, Beijing 100043, China; 3Department of Neonatology and NICU, Beijing Chao-Yang District Maternal and Child Healthcare Hospital, Beijing 100021, China; luck.2008sjz@163.com

**Keywords:** capillary leak syndrome, myocardial edema, newborn infant, cardiac ultrasound, point-of-care critical ultrasound (POC-CUS)

## Abstract

Capillary leak syndrome (CLS) is a rare, potentially life-threatening systemic disease with a mortality rate of more than 30%. Its major clinical manifestation and diagnostic basis are systemic hyperedema. However, we lack knowledge about the presence of severe myocardial edema in patients with CLS. If myocardial edema cannot be detected, it will become a dangerous hidden condition that threatens the safety of patient lives. With the routine application of point-of-care critical ultrasound (POC-CUS) in clinical practice, we found that 2 of 37 (5.41%) CLS patients had severe myocardial edema as the main manifestation. It is also necessary to distinguish it from myocardial noncompaction in newborn infants with severe myocardial edema. This paper will help us to have a deeper understanding and correct management of CLS and, thus, help us to improve the prognosis of patients. This article also suggests the necessity of routine implementation of POC-CUS in the neonatal intensive care unit.

This patient was a male infant, G_1_P_1_, born by emergency cesarean section due to severe fetal distress at a gestational age of 33^+1^ weeks. The birth weight was 2260 g, and the Apgar score was 1-7-10 points/at 1, 5 and 10 min of birth. He was admitted to the hospital due to dyspnea for 140 min after resuscitation from asphyxia. At birth, there was only a heartbeat, there was no spontaneous breathing and loss of muscle tone. The patient was transferred to our neonatal intensive care unit (NICU) after resuscitation with endotracheal intubation, positive pressure ventilation and chest compressions. Physical examination on admission showed that the premature infant had a poor appearance, poor mental response, hypotonia and disappearance of primitive reflexes. Arterial blood gas analysis showed pH 7.177, BE −16.6 mmol/L and a blood lactic acid value of 16.0 mmol/L. Severe disseminated intravascular coagulation (DIC) occurred rapidly after admission and was treated with heparin and low molecular dextrose. However, during the recovery stage of DIC, the patient gradually developed systemic highly nondepressed edema and significantly reduced urine volume and hypotension, and blood biochemical tests showed that the patient’s plasma albumin was significantly decreased. Based on these findings, we made a diagnosis of capillary leak syndrome (CLS). Point-of-care critical ultrasound (POC-CUS) examination showed that the infant had severe biventricular edema in addition to obvious lung edema. The ventricular wall of the infant was significantly thickened, the size of the heart chamber was significantly reduced with a reduction in ejection fraction and cardiac output. The thickness of the interventricular septum was 0.45 cm, the thickness of the left ventricular wall was 0.54 cm (Figure 1A), the thickness of the right ventricular wall was 0.55 cm and the diameter of the right ventricular outflow tract was 0.38 cm (Figure 1B). CLS was cured after 3 days of treatment with intravenous infusion of 3% NaCl (each dose of 3 ML/kg was administered intravenously over a 15–20 min period and repeated every 6 h) combined with furosemide (at a dose of 0.5 to 1.0 mg/kg per time,30 min after the intravenous injection of 3% NaCL). Repeat echocardiography showed that the thickness of the myocardial wall was significantly decreased and the inner diameter of the heart chamber was significantly increased after treatment. The inner diameter of the left ventricle was 1.62 cm (Figure 2A) and the inner diameter of the right ventricle was 0.81 cm (Figure 2B).

Capillary leak syndrome (CLS) is a rare, potentially life-threatening systemic disease caused by increased vascular permeability due to cytokine damage to the endothelium that causes the intravascular fluid and proteins to shift into the interstitial space, with subsequent hypovolemic hypotension and shock [1,2,3]. Generally, the major clinical characteristics of CLS include diffuse severe edema, hypovolemia or hypovolemic shock, hypoproteinemia and hemoconcentration [1,2,3]. With the routine application of POC-CUS in our clinical practice, we found that 2 of 37 (5.41%) CLS patients had severe myocardial edema as the main manifestation. This finding enriches our knowledge and deepens our understanding of CLS, thus contributing to its correct management and, finally, improving its prognosis. Another patient, also a male infant, G_2_P_2_, was born vaginally at 39 weeks of gestation with a birth weight of 2980 g. The infant had significant intrauterine distress (mainly presented as fetal heart rate slowed to 70–80 beats/min for 3 min) and mild asphyxia at birth (Apgar scores were 6 at 1 min of birth). The infant was admitted to our NICU 2 h after birth due to hyporesponsiveness. Physical examination on admission showed a full-term appearance and a slightly poor response. His vital signs were stable with a normal blood pressure. The muscle tone was slightly lower, and the primitive reflex slightly diminished. Routine POC-CUS examination showed marked thickening of the biventricular wall and narrowing of the cardiac chamber, especially in the right heart, with a small amount of pericardial effusion (Figure 3). Ventricular Noncompaction had once been suspected in this patient. However, the possibility of CLS could not be ruled out because of the high risk factors of CLS (hypoxia) and mild decreased in plasma albumin. Two days after intravenous injection of 3% NaCl combined with furosemide, echocardiography showed that myocardial thickness and cardiac chamber size had returned to normal.

The incidence of CLS has been reported to account for 1.62% of hospitalized critically ill neonates in the early stage [4], showing, however, an increasing trend: diffuse severe edema is often the first clinical manifestation observed by clinicians and the main basis for the diagnosis of CLS; myocardial edema as the main manifestation has not been described before. In recent years, with the routine implementation of POC-CUS in our NICU, we found severe myocardial edema in 2 of 37 (5.41%) CLS patients. Although the incidence of myocardial edema is not high, if it is not detected it may increase the mortality of children with CLS. Therefore, the clinical experience of these two cases suggests that echocardiography should be performed routinely in patients with high-risk factors for CLS to determine whether there is myocardial edema or not, so that the correct treatment can be given in a timely way to prevent a poor prognosis. 

The mortality rate of CLS has remained above 30% or even as high as 50% [4,5]. Our team has developed a combination of intravenous injection of 3% NaCl and furosemide to treat this disease with remarkable effectiveness. Nearly 40 patients were cured without complications such as electrolyte disturbances, showing the outstanding advantages of this treatment strategy.

It should be noted that severe myocardial edema could be misdiagnosed as ventricular noncompaction. Ventricular noncompaction mainly involved the left ventricle on echocardiography (92%), nearly half of the patients had a family history of ventricular noncompaction, echocardiography typically requires the following criteria: (1) the presence of multiple echocardiographic trabeculations, (2) multiple deep intertrabecular recesses communicating with the ventricular cavity, as demonstrated by color Doppler imaging and the recesses demonstrated in the apical or middle portion of the ventricle and (3) a 2-layered structure of the endocardium with a noncompacted to compacted ratio >1.4 [6,7,8,9,10]. In this study, case 2 had both left and right ventricle involvement but it was more severe in the right ventricle. According to our previous study, myocardial damage caused by asphyxia was mainly in the right heart [11]. This infant had a history including intrauterine and birth hypoxia, which is a common risk factor for CLS. Since this child also had mild hypoalbuminemia, we speculated that the infant could not be excluded from CLS caused by hypoxia. After we adopted specific treatment for CLS, the myocardial damage of the child returned to normal, allowing for the exclusion of myocardial noncompaction.

In conclusion, POC-CUS plays an increasingly important role in clinical practice, which not only contributes to the accurate diagnosis of diseases, but also helps to change the traditional understanding of diseases or increase the new understanding of diseases [12,13,14]. For these two patients, no different therapies were undertaken, and they had good outcomes after accepting the specific treatment to CLS. If there had been left ventricular outflow obstruction or reduced cardiac output, the therapeutic choice would have been much different. It is, therefore, important to evaluate the degree of left ventricular obstruction and any reduction in output of the patients with echocardiogram.

## Figures and Tables

**Figure 1 diagnostics-13-03597-f001:**
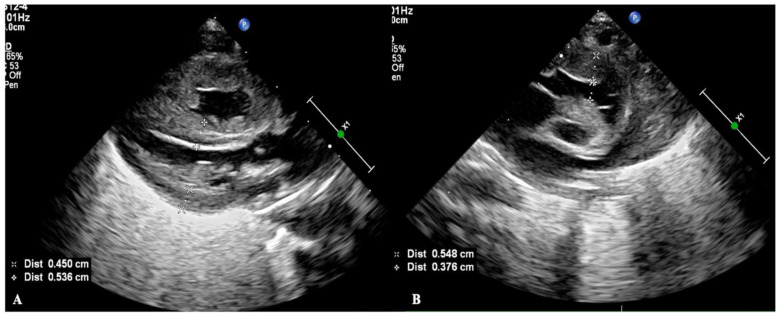
Myocardial edema in a CLS patient. POC-CUS showed the severe biventricular edema, the ventricular wall significantly thickened with the size of the heart chamber significantly reduced. (**A**) showed the thickness of the interventricular septum was 0.45 cm, the thickness of the left ventricular wall was 0.54 cm, (**B**) showed the thickness of the right ventricular wall was 0.55 cm, and the diameter of the right ventricular outflow tract was 0.38 cm.

**Figure 2 diagnostics-13-03597-f002:**
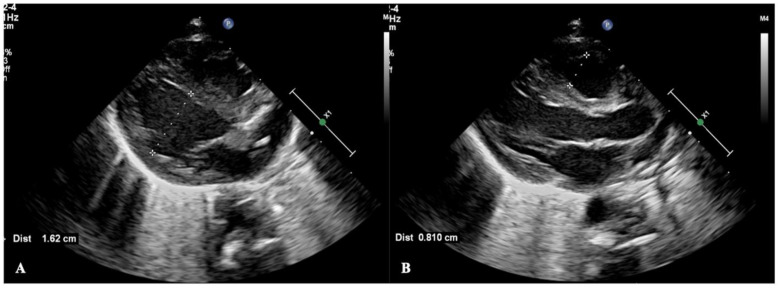
The myocardial edema disappeared after treatment. Echocardiography showed that the thickness of the myocardial wall significantly decreased, and the inner diameter of the heart chamber significantly increased after treatment. (**A**) showed the inner diameter of the left ventricle was 1.62 cm and (**B**) showed the inner diameter of the right ventricle was 0.81 cm at the end of diastole.

**Figure 3 diagnostics-13-03597-f003:**
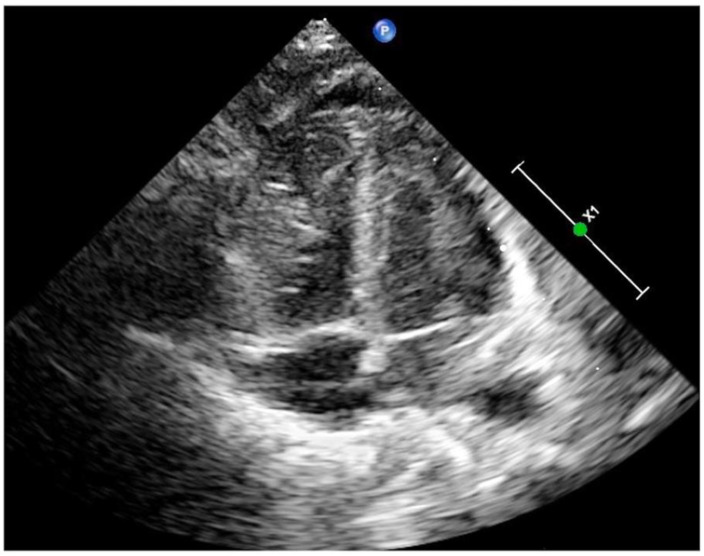
Echocardiographic image of case 2. Apex 4-chamber view of the heart showed a significant increase in the thickness of the whole myocardial wall and a significant reduction in the inner diameter of the heart chamber. The nonhomogenous and enhanced echo of the myocardium was once suspected as myocardial noncompaction.

## Data Availability

The dataset used and analyzed is available from the corresponding author upon reasonable request.

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
