# Peer review of "Myocardial Edema: A Rare but Specific Manifestation of Neonatal Capillary Leak Syndrome"

_diagnostics, 2023, doi:10.3390/diagnostics13233597_

Round 1

Reviewer 1 Report

Comments and Suggestions for Authors

The authors approach the role of POCUS for a pathology, capillary leak syndrome, critical condition with multiple organ failure. It is interesting point of view, and the presented images are suggestive.

The paper is well written. The term “ study” (road 94) should be changed, because this paper reports two cases.

Author Response

The authors approach the role of POCUS for a pathology, capillary leak syndrome, critical condition with multiple organ failure. It is interesting point of view, and the presented images are suggestive.

The paper is well written. The term “study” (road 94) should be changed, because this paper reports two cases.

Response: Dear reviewer, we agree with this comment. Please note we modified “the results of this study” to “the clinical experience of these two cases” in the revised edition.

-------------------------------------------------------------

Reviewer 2 Report

Comments and Suggestions for Authors

MINOR CONCERNS:

LINE 26 this is....was -wrong time sequence

LINE 43 biventricular

LINE 44 Specify the reduction in ejection fraction and whether cardiac output was reduced

LINE 49 specify the furosemide dose and time of administration. Specify the time of resolution.

FIGURE 2 specify the phase of the cardiac cycle (end  of diastole)

LINE 70 specify the parameters of intrauterine distress

LINE 75  biventricular

LINE 76 what about cardiac output. Was it preserved?

LINE 116 TO 122 I think the conclusions should be rewritten both for errors in form and repetition as prognosis-poor prognosis

MAIN CONCERNS:

LINE 107 the correct pediatric definition of noncompaction is a ratio, measurement in end- diastole ,greater than 1.4

In this regard, I would cite in the bibliography the article by Pignatelli et al:

Pignatelli RH, McMahon CJ, Dreyer WJ, et al. Clinical characterization of left ventricular noncompaction in children: a relatively common form of cardiomyopathy. Circulation 2003;108: 2672-8.

Another important consideration concerns the modification of therapy secondary to the finding of myocardial edema. the authors should specify that in their two patients no different therapy was undertaken but that if there had been left ventricular outflow obstruction or reduced cardiac output the therapeutic choice would have been different, and it is therefore important to evaluate with echocardiogram these patients , the degree of left ventricular obstruction and any reduction in output

Comments on the Quality of English Language

only minor changes are needed except the form of the conclusions

Author Response

LINE 26 this is....was -wrong time sequence.

Response: it has been revised.

LINE 43 biventricular.

Response: it has been revised.

LINE 44 Specify the reduction in ejection fraction and whether cardiac output was reduced.

Response: These conditions were present in the patient, and we modified them in the revised edition.

LINE 49 specify the furosemide dose and time of administration. Specify the time of resolution.

Response: We have added relevant information, please refer to the relevant content in parentheses in the revised manuscript.

FIGURE 2 specify the phase of the cardiac cycle (end of diastole).

Response: Yes, it was at the end of diastole. We have specified it in the revision.

LINE 70 specify the parameters of intrauterine distress.

Response: We added relevant information in the revised manuscript (mainly presented as fetal heart rate slowed to 70-80 beats/min for 3 minutes).

LINE 75  biventricular.

Response: It has been revised.

LINE 76 what about cardiac output. Was it preserved?

Response: We are sorry, no records in the medical records and ultrasound examination results.

LINE 116 TO 122 I think the conclusions should be rewritten both for errors in form and repetition as prognosis-poor prognosis.

Response: Dear reviewer, we have revised this part in the revised edition. We hope that the revised manuscript can express your opinion.

MAIN CONCERNS:

LINE 107 the correct pediatric definition of noncompaction is a ratio, measurement in end- diastole, greater than 1.4

In this regard, I would cite in the bibliography the article by Pignatelli et al: Pignatelli RH, McMahon CJ, Dreyer WJ, et al. Clinical characterization of left ventricular noncompaction in children: a relatively common form of cardiomyopathy. Circulation 2003;108: 2672-8.

Response: We thank you very much for the constructive suggestions. We have revised the article according to this literature.

Another important consideration concerns the modification of therapy secondary to the finding of myocardial edema. the authors should specify that in their two patients no different therapy was undertaken but that if there had been left ventricular outflow obstruction or reduced cardiac output the therapeutic choice would have been different, and it is therefore important to evaluate with echocardiogram these patients , the degree of left ventricular obstruction and any reduction in output

Response: Thank you very much for this constructive suggestion. We have revised this content as you suggested and hope it reflects your comments.